# Social Inequalities in Environmental Noise Exposure: A Review of Evidence in the WHO European Region

**DOI:** 10.3390/ijerph16061011

**Published:** 2019-03-20

**Authors:** Stefanie Dreger, Steffen Andreas Schüle, Lisa Karla Hilz, Gabriele Bolte

**Affiliations:** 1Department of Social Epidemiology, Institute of Public Health and Nursing Research, University of Bremen, 28359 Bremen, Germany; steffen.schuele@uni-bremen.de (S.A.S.); lisa-hilz@uni-bremen.de (L.K.H.); gabriele.bolte@uni-bremen.de (G.B.); 2Health Sciences Bremen, University of Bremen, 28359 Bremen, Germany

**Keywords:** environmental inequalities, environmental justice, noise, inequalities, systematic review, Europe, environment

## Abstract

Environmental noise is an important public health problem, being among the top environmental risks to health. The burden of noise exposure seems to be unequally distributed in societies. Up to now there is fragmentary evidence regarding which social groups are most affected. The aim of this review was to systematically assess published evidence on social inequalities in environmental noise exposure in the WHO European Region, taking different sociodemographic and socioeconomic dimensions as well as subjective and objective measures of environmental noise exposure into account. Articles published in English in a peer reviewed journal between 2010 and 2017 were included in the review. Eight studies were finally included in the review, four of them analysed aggregated data and four analysed individual data. Though results of social inequalities in noise exposures were mixed between and within studies, there was a trend that studies using indicators of material deprivation and deprivation indices showed higher environmental noise exposures in groups with lower socioeconomic position. More research on the social distribution of environmental noise exposure on a small spatial scale is needed, taking into account aspects of vulnerability and procedural justice.

## 1. Introduction

Environmental noise, defined as noise emitted from all sources except industrial workplaces [1], is an important public health problem. According to the World Health Organisation (WHO) noise is the top environmental risk to health [2]. It affects human health and well-being negatively and is a growing concern among both the general public and policy-makers in Europe. At least 100 million people in the European Union (EU) are exposed to road traffic noise levels that are classified as unacceptable by scientists and health experts. Alone in western Europe, at least 1.6 million healthy years of life are lost as a result of road traffic noise every year [2]. Effects of noise on health can be physiological as well as psychological [1,3,4]. As a direct injury to the auditory system, noise leads to auditory effects such as hearing loss and tinnitus. Noise also acts as a nonspecific stressor that has an adverse effect on human health, especially following long-term exposure. These health effects are the result of physiological and psychological distress, as well as a disturbance of the organism’s homeostasis and increasing allostatic load [3]. The EU Noise Directive [5], which is in place since 2002, aims to prevent and reduce environmental noise where necessary and to preserve environmental noise quality where it is good. However, the Directive does not address social inequalities in exposure to environmental noise.

From a social epidemiology perspective, noise may contribute to social health inequalities through an uneven distribution of exposure between socioeconomic groups [6,7]. Looking at exposure differences between social groups, studies suggest that some social groups are exposed to higher levels of environmental noise or have higher prevalences of noise exposure [8,9,10]. Social circumstances affect where and how people can afford to live, and socially deprived people tend to live and work in the lowest quality environments. As a consequence some socioeconomic groups may live in more polluted areas than other groups [11]. Relatively few studies investigate the distribution of noise exposure across different social groups in industrialized countries and the available evidence is not consistent [11]. Moreover, it is difficult to make general statements, because of great methodological differences between studies. Studies use different methods to assess noise exposure, different populations are investigated, and different social indicators are applied. Several studies, mainly conducted in Europe, suggest that more deprived people or communities are exposed to higher levels of traffic noise [10,12,13], or report higher annoyance due to noise [14]. Nonetheless, other studies provide counter-examples of high-income groups being exposed to higher levels of noise [13,15,16]. For example, a study conducted by Havard et al. [15] in France modelled annual average daily traffic in Paris. For each participant, exposure to road traffic noise within a 250m radius circular buffer centred on his/her exact residential building was assessed by averaging calculation points included within the buffer. After individual-level adjustment, participants’ noise exposure increased with neighbourhood educational level and dwelling value. The authors suggest that polluted city centre locations are often favoured by affluent groups, who avoid commuting [15]. Moreover, systematic analysis of data of the European Union Statistics on Income and Living Conditions (EU-SILC) from the years 2004–2009 on self-reported noise annoyance in terms of complaints about noise from neighbours or from the street, demonstrated differences within the WHO European Region: Social inequalities with a higher prevalence of noise exposure among individuals living in relative poverty were observed in most of the north-western European countries in contrast to no social differences or even lower exposure prevalences among low-income groups in south-eastern European countries [17].

Environmental health inequalities may not only arise as a result of exposure differentials, but also due to differences in vulnerability to the negative health effects of noise [11,18]. Chronic diseases or less healthy lifestyles may contribute to increased vulnerability to noise-related health effects. On the other hand, more affluent residents may have more resources to protect themselves against the negative effects of environmental noise, for instance by being able to afford better-constructed housing. Most likely, a combination of higher exposure, increased vulnerability, and less resources result in more pronounced noise-related health impacts among socially disadvantaged people [11]. This argument is underpinned by the fact that some studies indicate that more advantaged individuals are less likely to suffer from noise-related health impacts than poorer individuals, even if the advantaged individuals live in more noisy areas [11]. 

Vulnerability is an important aspect when analysing the health effects of environmental noise in different social groups. However, first of all systematic evidence on social inequalities in environmental noise exposure, systematically analysing different social dimensions, different measurement methods of noise and different study populations is needed. Therefore, the objective of this systematic review is to synthesise the evidence base on social inequalities in exposure to environmental noise in the WHO European Region. 

## 2. Materials and Methods

This systematic review has been registered with the PROSPERO international prospective register of systematic reviews database (Registration number: CRD42018099466) and was carried out following the PRISMA statement [19]. Social inequality was conceptualised with consideration of various sociodemographic and socioeconomic dimensions according to the PROGRESS-Plus framework [20].

### 2.1. Search Strategy

Studies were identified by searching the following electronic databases: MEDLINE (via Pubmed), SCOPUS and Web of Science on 23 April 2018. In accordance with the primary objective of this review, search terms included keywords for noise combined with terms for sociodemographic or socioeconomic characteristics and with terms for inequality/inequity or environmental justice using Boolean operators (see Table 1 and Appendix A). The term socioeconomic position (SEP) is used in this article for all facets of sociodemographic and socioeconomic characteristics. We base our understanding of SEP on the work of Krieger et al. [21,22] and expand the authors’ definition of SEP with other social characteristics (e.g., gender, age) that represent resource-based and prestige-based measures, which relatively position individuals, households, or neighbourhoods in society. Evidence obtained with this review will be part of an update of the report “Environmental Health Inequalities in Europe”, which has been published by the WHO Regional Office for Europe in 2012 [23]. Therefore, search results were restricted to articles published between 1 January 2010 and 31 December 2017. Additionally, the reference lists of all finally included articles were screened to capture any relevant publications missed by the electronic strategies mentioned above.

### 2.2. Eligibility Criteria

Original studies written in English and published in peer reviewed journals were included. As the results of the review will contribute to the above mentioned update of the WHO report, only articles conducted in the 53 Member States of the WHO European Region [24] were considered. Studies had to be quantitative observational studies (cohort, cross-sectional, or ecological). Qualitative studies as well as studies with a research focus on animals and their environment were excluded. To avoid excluding potentially disadvantaged populations, this review was not limited to a particular population but studies had to provide information about sociodemographic or socioeconomic characteristics of participants or study regions (measured at individual or aggregated level) according to any PROGRESS-Plus factor. We included studies which examined environmental noise exposures with motorized traffic as the main source. Studies examining occupational noise exposure only or examining noise from neighbours or leisure noise exposure were excluded. Measures of environmental noise included objectively measured noise parameters, proxies (e.g., traffic count data, distance to road), and subjective measures (e.g., noise annoyance). At least two population groups or study regions had to be compared concerning the extent of environmental noise exposure in relation to their socioeconomic or sociodemographic characteristics.

### 2.3. Data Collection and Synthesis

The database search was conducted by two reviewers independently. An EndNote database was created to store all retrieved records. After removing duplicates, all remaining titles and abstracts were screened independently by two reviewers to identify studies potentially meeting inclusion criteria. We calculated the Cohen’s Kappa for the inter-rater reliability agreement [25]. Full texts of potentially eligible studies were retrieved and systematically assessed for inclusion by one reviewer. The included studies were checked by a second reviewer. Any disagreements between the reviewers over the eligibility of particular studies were resolved by discussion and by consultation of two other reviewers. Additionally, the reference lists of all articles considered for data extraction were screened in order to capture potential relevant publications missed by the electronic database searches. For illustrating the study selection process, PRISMA guidelines [19] were used to produce a flow diagram. Also, the PRISMA checklist can be found in the supplements (Appendix A). Data from all included studies were extracted by one reviewer using a pre-designed and piloted extraction data form. Data extraction was checked by a second reviewer. Results of studies were only included in the review if numbers could be found. If results were only presented in a narrative way in the text, but specific numbers were not presented, these were not included in our analysis. The study results were systematically analysed according to the hypothesis, that socioeconomically deprived or socially disadvantaged individuals or areas have higher environmental burdens. In case of the social dimensions age and gender, old people, compared to other age groups, and females, compared to males, were regarded as disadvantaged groups.

Analyses were considered as descriptive analyses if no statistical test was performed to compare differences between groups. We compared the highest to the lowest SEP group in order to assess if environmental inequalities exist or not. Because operationalisations of SEP and noise exposure may be too heterogeneous, we did not define a specific cut-off for the magnitude of difference between SEP groups above which the differences were indicated as environmental inequalities. Analyses were defined as bivariate analyses if a statistical test was performed to test differences between groups. Three types of multivariate analyses were grouped together as multivariate analyses: (i) the SEP indicator analysed was actually the main independent variable analysed and the main research question and additional characteristics were considered as confounders, (ii) the SEP indicator was included in the model as a potential confounder variable for which the analysis was adjusted, (iii) several SEP indicators were included in the model simultaneously to reveal the influence of every single independent variable independently from all other factors, in the sense of mutual adjustment. Further, it is important to mention that with the aim of this review to assess social inequalities in environmental noise exposure the focus was put on bivariate analysis, as in multivariate analysis potential inequalities could be concealed by the inclusion of either several indicators of SEP or other variables in the analysis, which are not linked to our research question. In the data extraction sheet (see Appendix A) we extracted bibliographic details, the place where the study was conducted, the unit of analysis (analyses with individual data or aggregated data) and sample size, the study design, measurement and operationalisation of noise, the SEP analysed, the type of environmental inequality analyses and a summary of the result.

Results of the review were summarized in two different ways. First, we summarized the direction of social inequalities in environmental noise exposure by type of analyses of the included studies. Here, we also indicate the number of SEP indicators studied in each respective study. If in one study different SEP indicators were analysed in descriptive, bivariate, or multivariate analysis, or if different models were calculated or several operationalisations of environmental noise exposure were applied, then the results were summarized by one symbol for every type of analysis.

Second, we gave an overview over the different social inequalities in environmental noise exposure according to the different SEP indicators analysed. The references to the respective studies are also given in the table. Here, the thickness of the symbol indicates the type of analysis. Bivariate and multivariate analyses were grouped together. One symbol per type of analysis (descriptive or bivariate/multivariate) was included in the table. Because of the high amount of SEP indicators analysed, similar indicators were grouped together in the table (e.g., education deprivation, amount of highly educated people in the neighbourhood and individual education were grouped together as ‘education’). Moreover, if similar indicators are analysed in the same type of analyses (descriptive or bivariate/multivariate) they are represented in the table with one symbol only.

### 2.4. Risk of Bias

Due to the heterogeneity of study methods no quality assessment was conducted. All studies aimed to research noise exposure, but only half of the studies explicitly had the main research question to analyse social differences in noise exposure. Hence, a quality assessment would have assessed studies concerning a research question that was not the aim of the respective study. Furthermore, as our main research question primarily aimed to look at bivariate relationships, we were not able to apply a standardised quality assessment tool across studies.

## 3. Results

### 3.1. Description of Studies

Figure 1 shows the amended PRISMA flow diagram of study selection. The electronic database search identified 194 records. After duplicates were removed, 139 records were considered for title and abstract screening. The Cohen’s Kappa value for the inter-rater reliability agreement [25] was 0.55. Ten records were included into full text analysis, from which two were excluded. Eight studies [15,26,27,28,29,30,31,32] met all inclusion criteria and were finally taken into account for qualitative synthesis. No additional studies were identified by screening the reference lists of the eight included studies.

Table 2 provides an overview of the included studies. The majority of studies were conducted in France (*n* = 4) and Germany (*n* = 3), and one study was conducted in the UK. Most of the studies analyses road traffic noise. One study assessed noise annoyance due to road traffic noise, general transportation noise, and ambient noise [26]. One study used a noise indicator integrating different sources of traffic noise [30]. Of the eight studies included in the review, four used individual data [15,26,27,28] and four analysed aggregated data [29,30,31,32]. Of the studies with individual data, one study used a subjective operationalisation of noise exposure with questionnaires, one study applied an objective operationalisation of noise exposure based on noise maps, and two studies used both operationalisation types. All studies with individual data used single measures to assess SEP, one study additionally used a socioeconomic deprivation index (a single number calculated from an array of socioeconomic indicators). All studies with individual data performed multivariate analyses to assess environmental inequalities. In the majority of studies different SEP indicators were used in one model in the sense of mutual adjustment. In one study the SEP indicators were used as adjustment variables. One study additionally used descriptive analysis without performing a statistical test, and two studies additionally comprised bivariate analysis with a statistical test. Of the four studies with aggregated data, all used an objective operationalisation of noise. All studies with aggregated data used indices of socioeconomic deprivation to assess SEP and one study additionally used single indicators of SEP. Bivariate statistics were the most frequently applied method for analysing relations between indicators of SEP and environmental resources among the studies with aggregated data.

### 3.2. Associations between Sociodemographic and Socioeconomic Characteristics and Environmental Noise Exposure

Across all studies mixed results were found on how environmental noise exposure was linked to sociodemographic or socioeconomic characteristics of individuals or study regions (see Table 3 and Table 4). Studies where only one or a low number of SEP indicators were analysed, point towards a distribution of environmental noise to the disadvantage of people with lower SEP (see Table 3). Having a closer look at the single SEP indicators (see Table 4), opposing results were found for many single SEP indicators within and across studies. Indicators reflecting material aspects showed a trend of higher environmental noise exposure in low SEP-groups. For education, an indicator that not solely represents material circumstances but might also reflect behavioural aspects, contradicting results have been found. For age results suggest lower environmental noise exposure in older people. Here it has to be kept in mind that not age itself but other material factors linked to age might be associated with noise exposure. Two studies with aggregated data developed an index of social deprivation and applied it for one big city; both studies did not find significant results. The authors do not comment on whether the study had enough power to find relationships between indicators of SEP and environmental noise exposure. Two studies assessed noise annoyance and objective noise exposure based on noise maps. One of the studies linked the same indicators of SEP with noise annoyance as well as with objectively assessed noise [33]. Here, different indicators of SEP were associated with noise annoyance than with objectively measured noise. All studies analysing aggregated data assessed environmental inequalities for a big city or metropolitan area (see Appendix A). In all studies the individual or contextual SEP indicators were linked with noise exposure. However, not all of the studies had the main aim to specifically assess environmental inequalities. Four of the articles [15,28,29,31] (shortly) discuss mechanisms how environmental inequalities in environmental noise exposure could develop.

## 4. Discussion

### 4.1. Summary of Main Results

This review examined social inequalities in environmental noise exposure in the WHO European Region. Overall, the results of the included observational studies, which were conducted both on the ecological and individual level, were mixed between and within different indicators of SEP. Studies using indicators of material deprivation and studies using deprivation indices pointed towards higher environmental noise exposure in lower SEP groups. None of the studies found results pointing towards higher exposure in socially advantaged groups exclusively.

### 4.2. Social Inequalities in Environmental Noise Exposure

Results concerning environmental inequalities related to single SEP indicators were mixed and partly opposing results have been found. Also, because a broad range of SEP dimensions were analysed in a very limited number of studies, often one specific SEP dimension was examined in one study only, which makes a summary rather impossible.

Indicators representing material aspects, such as income, deprived living area, mean value of dwelling, or ownership of dwelling, point towards higher noise exposure in people with lower SEP. These material factors are associated to where people can afford to live.

For educational level, which not only represents material aspects, but might also be linked to behavioural aspects, opposing results have been found between studies. The study of Méline et al. [27] found opposing results for a “low proportion of highly educated residents” for different parameters of the noise exposure distribution: The conclusion whether social inequalities in noise exposure exist differed depending on whether the median, the 25th, or the 75th percentile of the noise exposure distribution was analysed. Thus opposing results concerning education within the study of Méline et al. [27] could be explained by different cut-offs for noise exposure in the analysis. In other analyses considering indicators measuring level of education at an area level, education was either not associated with noise exposure [32] or areas with a low proportion of highly educated residents had higher noise exposure in some analysis and a non-significant association in other analysis [27]. In studies focusing on individual educational level [27,28], individual low education was associated with higher noise exposure but not throughout all analyses. In the study of Riedel et al. [28], who analysed two different data sets in descriptive and multivariate analyses, individual low education was associated with higher noise exposure in descriptive analysis in one dataset, but not in the other analysis and not in the second dataset. It could be argued that educational level may have different meanings and importance in different countries and in urban or rural areas. However, the studies in this review which assessed education all analysed data from big cities or metropolitan areas and opposing results have been found in two studies from the same country and even similar country regions [15,27].

More recent studies, which were not part of our review, as they were published in 2018 or 2019 found small inequalities in environmental noise exposure. A study from London, UK, analysing socioeconomic and ethnic inequalities in exposure to noise pollution, reported that individuals with the highest household income, white ethnicity, and lowest income deprivation group were more likely to be exposed to aircraft noise. Participants in the group with the most area-level income deprivation were most likely to be exposed to rail noise [34]. A study from Ghent, Belgium, analysing residential exposure to noise found that only neighbourhoods with a higher percentage of people of a specific foreign origin (non-EU and non-Turkish-Maghreb) had a significantly higher exposure [35].

Moreover, if a study reported both significant and non-significant results of social inequalities in noise exposures, the overall trend of the analyses pointed towards inequalities and the non-significant results were mostly found in a sensitivity- or sub-analysis.

There might be a trend that indices of socioeconomic deprivation and SEP indicators representing material deprivation are associated with higher noise exposures. Moreover, migration status might be linked to higher noise exposure, too. However, due to the reasons mentioned above, results have to be interpreted with caution.

### 4.3. Comparison of Review Result with Evidence from Other Continents

Evidence from the WHO European Region based on the results of this review and the WHO report on inequalities in environmental noise exposure [17], and also older studies in the Region [9,10,16] suggest that a low SEP is associated with higher environmental noise exposure. The spatial scale of analysis, the operationalisation of SEP, and the operationalisation of noise are relevant factors influencing the association between SEP and environmental noise exposure. In the WHO European Region, there are differences between single countries where social groups tend to live. Often, the wealthier people live in the city centre, and people with lower SEP live in the suburbs [8]. However, this is not consistently the case in all countries of the Region.

In contrast to the WHO European Region, the proportion of people with a low SEP is often higher in the city centre and the more affluent tend to live in the more quiet suburbs in North America [8]. These considerations need to be kept in mind when comparing study results, especially if they are from different continents. In comparison to the WHO European Region a more consistent picture of inequalities in environmental noise exposure, often with a focus on ethnic minorities, can be found in North America: In Montreal, Canada, Carrier et al. [36] found that lower SEP at the neighbourhood level or a higher proportion of minority ethnicities was associated with higher noise levels. In another study in Montreal, Dale et al. [8] reported a strong correlation between noise exposure and different indicators representing socioeconomic disadvantage. However, given the inconsistency with a number of other studies, the authors suggest that the links between SEP and noise exposure are likely to be highly dependent on the local situation in each area studied. Therefore, a differentiated view with small scale analysis is necessary to gain valid results. A study in Minneapolis and St. Paul in the United States [12] reported a positive association between the level of road traffic noise and the proportion of individuals belonging to the non-white population (Asians, African-Americans, Hispanics), and living in low-income households. A study analysing estimated outdoor noise exposure in census block groups throughout the contiguous United States found evidence of higher noise exposure in census block groups with lower SEP and a higher proportion of Black, Hispanic, American Indian, and Asian residents [37]. These associations were stronger in more racially segregated communities. The more consistent results from North America, especially those analysing differences between ethnicities, could be due to a more pronounced segregation in U.S. cities compared to European cities [38]. Evidence from other continents is very rare. One exception is a study from Hong-Kong that quantified road traffic noise exposure of dwellings and related it to the SEP of the residents. The authors found a weak, but statistically significant association between lower income, lower educational attainment, and higher noise exposure at the street block level [39].

### 4.4. Vulnerability

The magnitude of exposure differences in different social groups may be relevant for health outcomes [3]. But not only the difference in exposure may lead to worse health in disadvantaged groups, but also an increased vulnerability [18]. One example is the study of Orban et al. in Germany, where among 3000 participants free from depressive symptoms at baseline, an annual average noise exposure of more than 55 dBA was associated with depressive symptoms after 5 years in the follow-up. However, this increase was only found in those participants with less than 13 years of education [40]. Evidence from studies suggest that deprived populations suffer worse health effects from noise through increased exposure and increased vulnerability to the effects of exposure, which is labelled as a ‘double burden’ [11]. A person’s resources and coping strategies may also reduce exposure to pollution and thus vulnerability. For example, Hajat et al. [41] comment that people from higher socioeconomic groups are more likely to be able to afford to live in better constructed houses with more effective noise insulation. Moreover, they have more social capital, which, in this case, may be political influence used to prevent polluting land uses, such as big roads and railway ways being built in the local community. Also, more affluent people might have more resources to protect themselves against environmental (noise) exposure by working indoors or paying for better-insulated windows [41]. A study from Rome reported that the more affluent people live in the city centre but have an additional country estate for recreation at the weekend or during vacations. As a result, they are not exposed to the actual levels of noise at their first residence all the time [42].

### 4.5. Implication for Practice

Notwithstanding these reported uncertainties, there is evidence that enables action in practice and policies. Indisputably, reducing noise will have a positive impact on health for all people. In addition to widespread actions, targeted measures aimed to reduce noise exposure in socioeconomically deprived populations will reduce the risk of society’s poorest to suffer from greater health consequences than the more affluent population related to noise pollution [11]. Spatial planning to identify priority areas for noise reduction, building design, and protection and enhancement of quiet areas may play a role in improving living conditions and pollution levels for socioeconomically deprived groups [43]. Aside from that, action to promote and adopt more sustainable forms of transport can have benefits for all population groups [11]. By this, a reduction of noise for all parts of society and targeted measures for the most at-risk groups in the sense of proportionate universalism could be achieved [44]. Also, targeted measures to reduce the vulnerability of deprived populations to the health effects of noise exposure should be taken to warrant that these populations are not at greater risk due to higher exposure, increased vulnerability, and less resources. For example Riedel et al. [45] suggest that dwelling-related environmental resources, such as having access to green space and a quiet side, decrease noise annoyance in the elderly. This is in line with the European Environment Agency, which highlights the need to preserve ‘environmental noise quality where it is good’, as well as to preserve quiet areas [46]. Apart from noise action planning, urban morphology can have a positive effect on the reduction of noise and thus might help to decrease inequalities in environmental noise exposure [47]. Margaritis and Kang highlight that green spaces, building and road attributes impact on noise distribution in cities [47]. These elements can be targeted in an integrative approach when aiming to reduce noise exposure of populations. Also, considering noise and health research more comprehensively, Lekaviciute et al. [48] suggest that noise mapping methods need to be harmonised across different countries. Moreover, new methods should be designed to measure total noise exposure—from more than one source—and to separate exposures from different sources, such as traffic and industry [48]. The decibel measurements used in most studies do not differentiate between noise sources. Another important aspect is that noise action planning should be designed in a way that social inequalities should not be caused or increased [33]. The EU noise directive [5] does not consider aspects of vulnerability or procedural justice, which are two aspects that are important to address in future versions of the Directive.

### 4.6. Implications for Further Research

There are relatively few studies focusing on the distribution of noise exposure between different social groups. In some of the included studies, assessing social inequalities in noise exposure was not the main research question, but numbers were found in sub-analyses or the SEP dimension was used as an adjustment variable. Therefore, studies specifically designed to investigate social inequalities in environmental noise exposure are urgently needed. Included studies and most other studies on noise exposure assess noise exposure at one exposure point (e.g., at home) only. Multiple exposures at different places (e.g., at home, at work, during leisure time activities) were not assessed in any of the included studies. This fact could be of particular importance when looking at social inequalities in noise exposure, as socially deprived people may not only live in noisier areas but also work in jobs with elevated noise exposure. This fact can lead to an underestimation of social inequalities in noise exposure. Also, ideally studies should not only include noise exposure at home, but also when and how much time people actually spent at home, which again could differ greatly in different social groups. Studies in this review used subjective and objective measures to assess environmental noise exposure. However, it is important to mention that subjective measures and objective measures do not always correlate to a high degree and, moreover, are not necessarily linked to the same socioeconomic indicators, which can be seen in the study of Riedel et al. [28]. Here, objective noise exposure was higher in people with younger age and people with lower education in descriptive analysis. In multivariate analysis, these factors were not linked to subjective noise annoyance, but migration was associated with higher noise annoyance in one of the analysed datasets in this study. Moreover, a study conducted in Dortmund, Germany, found that individuals with a low SEP were more likely to feel subjectively annoyed by the same magnitude of objective environmental traffic noise exposure than individuals with a high SEP [49]. As a result, environmental inequalities assessed with objective measures could potentially underestimate the health impact of inequalities between low and high SEP people. People with a low SEP could be more sensitive to the same extent of environmental noise exposure than people with a high SEP. Therefore, objective and subjective measures should not be used interchangeably, especially when environmental health inequalities are analysed. Another challenge for future research is to estimate how much difference in noise exposure may explain social disparities in noise related health outcomes, or the other way how much of health inequalities in noise related health outcomes can be attributed to differences in noise exposure taking vulnerability of social groups into account. In this context longitudinal studies are of high importance.

### 4.7. Strengths and Limitations

The results of this review may be affected by several limitations. First, the results of this review are vulnerable to publication bias. Second, the review only considered studies published in English and from WHO European Region countries, therefore, a generalization to other cultures or countries might be limited. The focus on WHO European Region countries is due to the fact that results of this review are part of monitoring activities within Europe, and contribute to a wider picture together with a report of the WHO Regional Office for Europe [23], which will be updated in 2019, in which survey data are systematically analysed.

As said before, the focus of this review was on bivariate analyses as multivariate analyses might mask inequalities. A systematic comparison between bivariate and multivariate results was not possible as the single studies included different SEP indicators. Our review visualized the directions of associations between SEP indicators and measures of environmental noise, and if associations were statistically significant or not. The included studies applied different statistical methods (e.g., description vs. correlation vs. regression), different operationalisations of SEP characteristics (e.g., continuous variables vs. categorical variables with different numbers of categories), and different reference categories for single indicators were used. Therefore, an analysis estimating the magnitude of inequalities was not possible. Only a few studies gave information on aspects of mechanisms of the development of inequalities in environmental noise exposure (e.g., procedural justice). To be able to comprehensively assess mechanisms leading to social inequalities in environmental noise exposure, a specific systematic review aiming at that question would be necessary. Due to the heterogeneity of study methods no quality assessment was conducted. For the majority of included studies the analysis of environmental inequalities in noise exposure was not the main research question. Hence, the quality assessment would have assessed studies concerning a research question that was not the aim of the respective study. Because of the reasons stated above a quality assessment would not have contributed to the evidence of this review.

A strength of this review is that we included different measures of environmental noise exposures in the search. Moreover, we included a broad range of SEP indicators measuring different aspects of sociodemographic and socioeconomic stratification [50,51]. By this, we were able to analyse whether the type of noise assessment (subjective or objective) and whether specific SEP indicators were relevant for inequalities in environmental noise exposure. Unfortunately, there were not enough studies analysing different dimensions of SEP, which impeded comparability of studies.

Notwithstanding the above mentioned limitations, the present review provides an important contribution to public health research. The review offers a systematic analysis of social inequalities in environmental noise exposure in the WHO European Region that includes studies with individual and aggregated data.

## 5. Conclusions

In conclusion, this review summarizes the evidence on social inequalities in environmental noise exposure in countries of the WHO European Region. The included studies and their methods are very heterogenic; furthermore, a high number of different SEP indicators was covered by the included studies. A trend might be observed that general indices of deprivation and social indicators at the individual level representing material aspects are associated with higher environmental noise exposure. The number of included studies per social indicator is too low to draw general conclusions for individual SEP indicators. Therefore, more research on the social distribution of environmental noise exposure is urgently needed. Monitoring of social inequalities in environmental noise exposure on a small spatial scale should be an integral part of structural health monitoring activities. Based on that, evidence-based preventive measures against environmental noise exposure could be developed and groups that have the highest exposure and are potentially more vulnerable could be specifically targeted.

## Figures and Tables

**Figure 1 ijerph-16-01011-f001:**
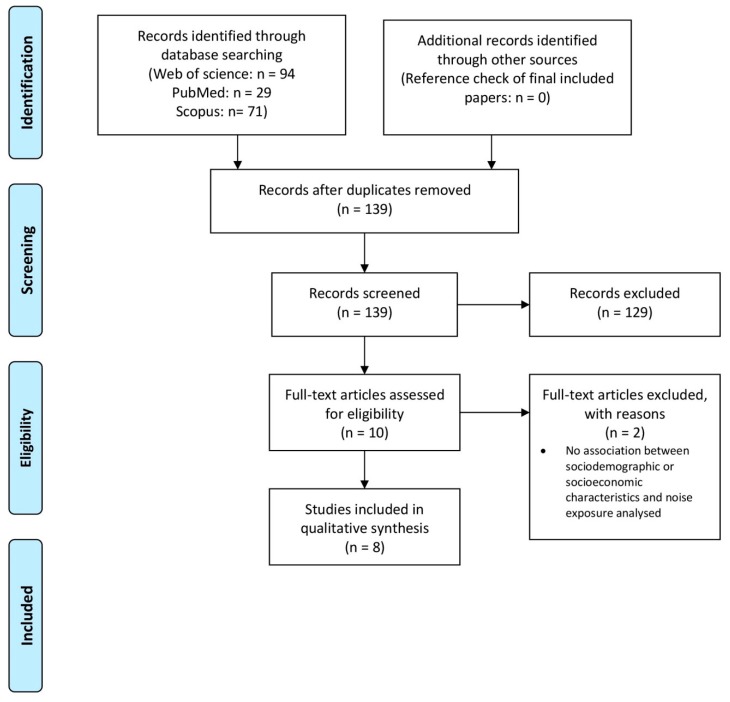
Flow Diagram of included studies [19].

**Table 1 ijerph-16-01011-t001:** Search terms and Medical Subject headings in PubMed.

Search	Query
#1	(“sociological factors” [MeSH Terms] OR disadvantaged [All Fields] OR disadvantage [All Fields] OR deprived [All Fields] OR social [All Fields] OR socio*[All Fields] OR “vulnerable populations” [MeSH Terms] OR vulnerable [All Fields] OR vulnerability [ALL Fields] OR psychosocial [All Fields] OR psycho-social [All Fields] OR “socioeconomic factors” [MeSH Terms] OR socio-economic [ALL Fields] OR deprivation [All Fields] OR socio-demographic [All Fields])
#2	(“noise, transportation” [MeSH Terms] OR noise [Title/Abstract])
#3	(inequality [Title/Abstract] OR inequity [Title/Abstract] OR inequities [Title/Abstract] OR inequalities [Title/Abstract] OR unequal [Title/Abstract] OR “environmental justice” [Title/Abstract] OR “environmental injustice” [Title/Abstract])
#4	(“2010/01/01” [Date–Publication]: “2017/12/31”[Date–Publication])
Final search	#1 AND #2 AND #3 AND #4
Filters selected manually	Language: English Species: Humans

**Table 2 ijerph-16-01011-t002:** Characteristics of studies on social inequalities in environmental noise exposure.

Author, Year	Noise Operationalisation		SEP Measures		Type of Environmental Inequality Analysis			Country
	Subjective	Objective	Single Measure	Index	Descriptive	Bivariate	Multivariate	
**Studies with individual data**								
Grelat et al., 2016 [27]	x		x				x	France
Havard et al., 2012 [15]		x	x	x		x	x	France
Méline et al., 2013 [28]	x	x	x		x		x	France
Riedel et al., 2014 [29]	x	x	x			x	x	Germany
**Studies with aggregated data**								
Xie and Kang, 2010 [33]		x	x	x		x		UK
Lakes et al., 2014 [32]		x		x		x		Germany
Flacke et al., 2016 [31]		x		x		x		Germany
Bocquier et al., 2012 [30]		x		x		x		France

SEP: socioeconomic position.

**Table 3 ijerph-16-01011-t003:** Summary of relationships between SEP indicators and (prevalences of) environmental noise exposure across studies.

Author/Study	Descriptive		Bivariate		Multivariate	
	Number of SEP Indicators in the Study	Relationship	Number of SEP Indicators in the Study	Relationship	Number of SEP Indicators in the Study	Relationship
**Studies with individual data**						
Grelat et al., 2016 [27]		-		-	3	↑
Havard et al., 2011 [15]		-	3	↕	7	↕
Méline et al., 2013 [28]	1	↕		-	6	↕
Riedel et al., 2014 [29]		-	4	↕	4	↑
**Studies with aggregated data**						
Bocquier et al., 2012 [30]		-	1	↑		-
Flacke et al., 2016 [31]		-	1	n.s.		-
Lakes et al., 2014 [32]		-	1	n.s.		-
Xie and Kang, 2010 [33]		-	8	↕		-

↑: low SEP groups have higher noise exposure or higher prevalences of environmental noise exposure compared to high SEP groups or higher SEP groups have lower environmental noise exposure or lower prevalences of environmental noise exposure compared to lower SEP groups (Relation for one or more social dimensions found, plus potentially non-significant results for some SEP indicators). ↕: Both directions found/Opposing relations found within the study with different SEP indicators or with different operationalisations of noise. -: not reported in the study. n.s.: no significant inequalities in environmental noise exposure found. Female and old age are regarded as categories of disadvantage. Only one symbol was given to summarize the results of one study across all SEP indicators.

**Table 4 ijerph-16-01011-t004:** Social inequalities in noise exposure by SEP indicator and type of analyses.

SEP Indicator	Analyses with Individual Data	Analyses with Aggregated Data
High deprivation (indices) [27,30,31,32,33]	**↑**	**↑↑ - -**
Deprived living area [28,33]		**↑**
Low income [28,33]	**↑**	**↓ ^a^**
Low mean value of dwelling [15]	**↓^a^**	
Non-Ownership of dwelling [28]	**↑**	
100% social housing in neighbourhood [27]	**↑**	
Employment deprivation [33]		**-**
Low % of households without a car [27]	**↑**	
Low education [15,28,29,33]	**↓^a^ ↕ ↑ -**	**-**
High % of people being disabled [33]		**-**
Old age [28,29,33]	**↓^a^ -**	**-**
Gender (male as reference) [28,29]	**- -**	
Migration [15,29]	**↑^a^ ↑^a^**	
Low human development index [15]	**↓^a^**	

**↑** Bivariate/Multivariate Analysis: low SEP groups have higher (prevalences of) environmental noise exposure compared to high SEP groups or high SEP groups have lower (prevalences of) environmental noise exposure compared to low SEP groups (Relation for one or more social dimensions found). **↓** Bivariate/Multivariate Analysis: low SEP groups have lower (prevalences of) environmental noise exposure compared to high SEP groups or high SEP groups have higher (prevalences of) environmental noise exposure compared to low SEP groups. **↕** Bivariate/Multivariate Analysis:contrasting results within study (different operationalisations for noise were analysed within studies). **-** Bivariate/Multivariate Analysis: no social disadvantage found **^a^** Sig-nificant association not found for all operationalization of noise or all statistical analyses. One symbol per type of analysis (descriptive or bivariate/multivariate) was included in the table.

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
