# Peer review of "Social Inequalities in Environmental Noise Exposure: A Review of Evidence in the WHO European Region"

_ijerph, 2019, doi:10.3390/ijerph16061011_

Round 1

Reviewer 1 Report

The authors have presented a sound methodology, which is a crucial aspect in such extensive literature reviews. There are a few more aspects that need further clarification as regards the filters they used to end up in the number of included studies presented in Figure 1, thus these issues can be addressed with minor revisions. This review can be a milestone for further studies on the field of traffic noise and social inequalities, although perceptual studies were not considered.

detailed comments:

Reviewer’s comments

The authors have presented a sound methodology, which is a crucial aspect in such extensive literature reviews. There are a few more aspects that need further clarification as regards the filters they used to end up in the number of included studies presented in Figure 1, thus these issues can be addressed with minor revisions. This review can be a milestone for further studies on the field of traffic noise and social inequalities, although perceptual studies were not considered. Detailed comments are presented below:

L.142: please refer also in the text the total number of studies finally being considered. How many of these studies had noise exposure as the main research question. In Lines 184 the word “majority” is mentioned, but you’d better be more precise.

L.199: from Table 2 it seems that you excluded 129 out of 139 studies. Please make clear whether the reason was that they were not directly relevant to noise or something else.

L.282: “there is at trend…”. It is difficult to identify a trend with such a limited number of case studies. In how many of them was this “trend” consistent?

L.312: “in most cases” , please mention the exact number.

L.359: “Evidence from studies”, is it just one study or more?

L.374: “targeted measures”, please be specific.

L.377 - 379: I found this sentence quite “generic”. Can you please be more specific by mentioning some details from Noise Action Plans or the Good Practice Guide on quiet Areas  (EEA, 2014)?

L.383: “total noise exposure”, please clarify this referring to the page number of reference 45. In the way it is written there might be a confusion with Lden. What is mentioned in  this document (ref 45, p.82) is the need for “proxy measures  of noise exposure”.

L.386: apart from noise action planning, urban morphology is another parameter that can have a positive effect on traffic noise reduction and therefore on social inequalities, as outlined by Margaritis and Kang (2016) in the article “Relationship between urban green spaces and other features of urban morphology with traffic noise distribution”.

L.387-88: social inequalities usually pre-exist and the aim of noise action planning is to blunt or minimize them.

L.413 - 416: this part refers to two different approaches (objective, subjective). Do you actually suggest the integration of both approaches in future case studies when you mention the issue of “interchangeability”?

Overall there is a big amount of literature on “subjective noise annoyance” and environmental noise. Why this part of literature was not considered in this research, which could potentially level up the final number of included studies?

L.440: there are inherent causes of social inequalities such as the level of education, housing parameters, health, etc. which have already been mentioned to some extent in this research. What more do you expect to find out with another review?

L.446: “different measures of environmental noise exposure”. Please be more specific.

L.462: please mention the specific number of included case studies in parenthesis [eg. “too low (8)”].

Author Response

Reviewer 1

The authors have presented a sound methodology, which is a crucial aspect in such extensive literature reviews. There are a few more aspects that need further clarification as regards the filters they used to end up in the number of included studies presented in Figure 1, thus these issues can be addressed with minor revisions. This review can be a milestone for further studies on the field of traffic noise and social inequalities, although perceptual studies were not considered.

Response: We would like to thank reviewer 1 for his/her comments and his/her positive feedback.

detailed comments:

Reviewer’s comments

The authors have presented a sound methodology, which is a crucial aspect in such extensive literature reviews. There are a few more aspects that need further clarification as regards the filters they used to end up in the number of included studies presented in Figure 1, thus these issues can be addressed with minor revisions. This review can be a milestone for further studies on the field of traffic noise and social inequalities, although perceptual studies were not considered. Detailed comments are presented below:

L.142: please refer also in the text the total number of studies finally being considered. How many of these studies had noise exposure as the main research question. In Lines 184 the word “majority” is mentioned, but you’d better be more precise.

Response: Please see lines 204-205, where we state “Eight studies met all inclusion criteria and were finally taken into account for qualitative synthesis.”

All studies had noise exposure as the main research question; however, not all of them had the main research question to analyse social differences in noise exposure. We adapted the manuscript to make this point more clear.

L.199: from Table 2 it seems that you excluded 129 out of 139 studies. Please make clear whether the reason was that they were not directly relevant to noise or something else.

Response: We performed Title and Abstract screening on 139 studies and excluded 129 because they did not meet our inclusion criteria, which are described under 2.2 Eligibility criteria (please see line 115-130). In lines 204-205 we state in the manuscript “Eight studies met all inclusion criteria and were finally taken into account for qualitative synthesis.”

L.282: “there is at trend…”. It is difficult to identify a trend with such a limited number of case studies. In how many of them was this “trend” consistent?

Response: Thank you for this valuable comment. We reworded the sentence in the manuscript to adapt it to the situation of a small number of studies (see line 353).  

L.312: “in most cases” , please mention the exact number.

Response: Thank you for this comment. In fact this is true for all cases (two studies); we adapted that in the manuscript (see lines 385-386).

L.359: “Evidence from studies”, is it just one study or more?

Response: The reference given here is an in depth-report produced for the European Commission, which include results of a large number of studies.

L.374: “targeted measures”, please be specific.

Response: We integrated this comment into the manuscript and now mention specific examples of interventions here (see lines 463-473).

L.377 - 379: I found this sentence quite “generic”. Can you please be more specific by mentioning some details from Noise Action Plans or the Good Practice Guide on quiet Areas  (EEA, 2014)?

Response: We integrated this comment into the manuscript and now mention specific actions that could be taken here.

L.383: “total noise exposure”, please clarify this referring to the page number of reference 45. In the way it is written there might be a confusion with Lden. What is mentioned in this document (ref 45, p.82) is the need for “proxy measures  of noise exposure”.

Response: On page 57 in this document (reference 45) it is said: “A future challenge is to develop methods to assess the total noise exposure and to disentangle the effects from different sources.”

As we explicitly state “from more than one source” in the sentence in line 475, we do think that a confusion with Lden is unlikely.

L.386: apart from noise action planning, urban morphology is another parameter that can have a positive effect on traffic noise reduction and therefore on social inequalities, as outlined by Margaritis and Kang (2016) in the article “Relationship between urban green spaces and other features of urban morphology with traffic noise distribution”.

Response: Thank you very much for this valuable input. We integrated this point in our manuscript and reference the study of Margaritis and Kang.

L.387-88: social inequalities usually pre-exist and the aim of noise action planning is to blunt or minimize them.

Response: We do agree with the statement that social inequalities usually pre-exist. However, noise action planning and the EU noise directive do not explicitly have the aim to reduce social inequalities, but focus on a general decrease of noise exposure of the population. With the sentence you mention (now LL 478-479) we highlight, that if interventions to reduce noise exposure are planned, it is important to design them in a way that no so called intervention-generated inequalities develop. This can happen if an intervention is of greater benefit to advantaged (lower-risk) groups than to disadvantaged (higher-risk) groups.

L.413 - 416: this part refers to two different approaches (objective, subjective). Do you actually suggest the integration of both approaches in future case studies when you mention the issue of “interchangeability”?

Response: Thank you for this comment. We do not think that in future research subjective and objective measures of noise should necessarily be considered. We highlight the point that research has shown that socioeconomic deprived individuals show higher subjective annoyance given the same objective exposure compared to individuals of higher socioeconomic position. This shows that an assessment of objective exposure does not necessarily reflect the subjective annoyance.

Overall there is a big amount of literature on “subjective noise annoyance” and environmental noise. Why this part of literature was not considered in this research, which could potentially level up the final number of included studies?

Response: We did include literature on subjective noise annoyance. Please see line 127-128, where we state: “Measures of environmental noise included objectively measured noise parameters, proxies (e.g. traffic count data, distance to road), and subjective measures (e.g. noise annoyance).”

L.440: there are inherent causes of social inequalities such as the level of education, housing parameters, health, etc. which have already been mentioned to some extent in this research. What more do you expect to find out with another review?

Response: Thank you for this comment. In this sentence we mention that our review did not address mechanisms that lead to environmental inequalities, as this was not the aim of our review. We focus on the description of which socioeconomic indicators are linked to environmental inequalities. We do not address how the single socioeconomic indicators produce environmental inequalities. This question could be systematically addressed by a review.

L.446: “different measures of environmental noise exposure”. Please be more specific.

Response: More information on the different measures potentially included is given in the methods section. Please see ll 127-128, where we state: “Measures of environmental noise included objectively measured noise parameters, proxies (e.g. traffic count data, distance to road), and subjective measures (e.g. noise annoyance).”

L.462: please mention the specific number of included case studies in parenthesis [eg. “too low (8)”].

Response: In table 4 the single socioeconomic indicators are displayed. There it becomes obvious that many socioeconomic indicators are only analysed in one (e.g. low mean value of dwellings, non-ownership of dwellings, 100% social housing in neighbourhood) or two (e.g. gender, low income) studies. As this information is given in the results section of the paper, we decided to not put the specific numbers in the discussion section, as also there is no clear cut-off that can be regarded as “enough” to make generalized statements.

Reviewer 2 Report

General comments: 

Overall the paper was very interesting to read and presents some important findings regarding the role of context-specific expectations of residents regarding sound and annoyance. The findings also highlight the need for more nuanced approaches to predicting noise annoyance beyond demographic stereotypes and how urban living trends can modify the perceptions of those affected by noise. 

The methodology was rigorous and presented in a clear and thorough way. 

Specific comments:

L13 - insert "regarding between "evidence" and "which"

L 46-48 - Sentence starting with "Looking at ... noise exposure." This statement needs supporting references.

L145 - rather than "old people" state age/age range instead

L444 - "albeit because" => "because of the"

L459 - "was" => "were"

Author Response

Reviewer 2

General comments: 

Overall the paper was very interesting to read and presents some important findings regarding the role of context-specific expectations of residents regarding sound and annoyance. The findings also highlight the need for more nuanced approaches to predicting noise annoyance beyond demographic stereotypes and how urban living trends can modify the perceptions of those affected by noise. 

The methodology was rigorous and presented in a clear and thorough way. 

Response: We would like to thank reviewer 2 for his/her comments and his/her positive feedback.

Specific comments:

L13 - insert "regarding between "evidence" and "which"

Response: Thank you for this comment. We adjusted it in the manuscript.

L 46-48 - Sentence starting with "Looking at ... noise exposure." This statement needs supporting references.

Response:  We added supporting reference in the manuscript – see line 48.

L145 - rather than "old people" state age/age range instead

Response:  Thank you very much for this comment. As age is operationalized and categorized differently in the studies (e.g. linear, different categorization of age groups) it is not possible to give an age or age range here.

L444 - "albeit because" => "because of the"

Response:  Thank you for this comment; we changed that accordingly in the manuscript.

L459 - "was" => "were"

Response:  We are not exactly sure which sentence you are referring to, as the line number does not really match. If you mean the sentence in LL 554-556 “The included studies and their methods are very heterogenic; furthermore, a high number of different SEP indicators was covered by the included studies.” Then the ”was” is linked to “a high number” – therefore we think it needs to be “was” not “were”.

Reviewer 3 Report

The   issue addressed by the article is quite interesting in that the problem of   noise in major cities is a daily occurrence for all who live in them and is   potentially a "hidden" health hazard. To combine the nuisance of   noise with social inequalities increases the research interest, despite the   reasonable difficulties that it presents mainly in searching for appropriate   data from different sources.

On   a more general level, the authors have given to the article the required   scientific documentation and completeness of bibliographic sufficiency. They   also follow a clear and documented structure by taking an objectively sound   scientific methodology to reach their specific goal.

The   authors actually use a wide range of different data from different   heterogeneous sources a reason very likely to hamper their work. But the method   they followed in order to approach the information they needed through   structured queries that they put into corresponding databases was very apt. I   would even say that the wonderful knowledge they have in this specific   subject of computer science has given me a very positive impression, despite   the fact that they come mainly from another field of knowledge. This shows   that they have particularly grasped the value of information retrieval,   through informatics which highlights the interdisciplinary dimension of their   research.

Overall,   I would say that this article would be able to be published immediately, but   I do believe that writers will have to bother to correct some of the   expressive failures I noticed by reading, which make it difficult to be the   meaning understandable.

In   particular (lines 54 to 55), in the phrase "different populations are investigated   and different social indicators are applied. ..." I think that the   adverb "as" after the word "exposure" is missing so as to   link the two sentences.

Also,   the text included between lines 64 to 71, I think it needs to be considered   more, as tricky. I believe that the meaning that writers want to give is not   right.

For   these reasons alone, I believe that the article may be published with minor   revision.

Author Response

Reviewer 3

The   issue addressed by the article is quite interesting in that the problem of   noise in major cities is a daily occurrence for all who live in them and is   potentially a "hidden" health hazard. To combine the nuisance of   noise with social inequalities increases the research interest, despite the   reasonable difficulties that it presents mainly in searching for appropriate   data from different sources.

On   a more general level, the authors have given to the article the required   scientific documentation and completeness of bibliographic sufficiency. They   also follow a clear and documented structure by taking an objectively sound   scientific methodology to reach their specific goal.

The   authors actually use a wide range of different data from different   heterogeneous sources a reason very likely to hamper their work. But the method   they followed in order to approach the information they needed through   structured queries that they put into corresponding databases was very apt. I   would even say that the wonderful knowledge they have in this specific   subject of computer science has given me a very positive impression, despite   the fact that they come mainly from another field of knowledge. This shows   that they have particularly grasped the value of information retrieval,   through informatics which highlights the interdisciplinary dimension of their   research.

Overall,   I would say that this article would be able to be published immediately, but   I do believe that writers will have to bother to correct some of the   expressive failures I noticed by reading, which make it difficult to be the   meaning understandable.

Response: We would like to thank reviewer 3 for his/her comments and his/her very positive feedback.

In   particular (lines 54 to 55), in the phrase "different populations are investigated   and different social indicators are applied. ..." I think that the   adverb "as" after the word "exposure" is missing so as to   link the two sentences.

Response: This sentence is an enumeration of methodological differences between studies. In fact “different methods to assess noise exposure” and  “different populations are investigated” are to separate reasons. The inclusion of the adverb “as” would link these two reasons in a way that would change the intended meaning of our sentence.

Also,   the text included between lines 64 to 71, I think it needs to be considered   more, as tricky. I believe that the meaning that writers want to give is not   right.

Response: Thank you for this comment. We adapted the manuscript so that it becomes more clear that we speak of two different studies here.

For   these reasons alone, I believe that the article may be published with minor   revision.
